# Regional characterization of meteorological and agricultural drought in Baluchistan province, Pakistan

**Muhammad Rafiq**[1], **Yue Cong Li**[1]\*, **Ghani Rahman**[2,3]\*, **Khawar Sohail**[4], **Kamil Khan**[5], **Aun Zahoor**[4], **Farrukh Gujjar**[4], **Hyun-Han Kwon**[2]

**1** Key Laboratory of Environmental Evolution and Ecological Construction of Hebei Province, Hebei Normal University, Shijiazhuang, PR China, **2** Department of Civil and Environmental Engineering, Sejong University, Seoul, Republic of Korea, **3** Department of Geography, University of Gujrat, Punjab, Pakistan, **4** Geological Survey of Pakistan Quetta, Baluchistan, Pakistan, **5** Department of Seismology and Geophysics Studies, University of Baluchistan, Baluchistan, Pakistan

\* ghani.rahman@sejong.ac.kr (GR); lyczhli@aliyun.com (YCL)

## Abstract

Drought is a complex natural hazard that occurs when a region experiences a prolonged period of dry conditions, leading to water scarcity and negative impacts on the environment. This study analyzed the recurrence of drought and wet spells in Baluchistan province, Pakistan. Reconnaissance Drought Index (RDI), Standardized Precipitation Evapotranspiration Index (SPEI), and Vegetation Condition Index (VCI) were used to analyze droughts in Baluchistan during 1986–2021. Statistical analysis i.e. run theory, linear regression, and correlation coefficient were used to quantify the trend and relationship between meteorological (RDI, SPEI) and agricultural (VCI) droughts. The meteorological drought indices (1, 3, 6, and 12-month RDI and SPEI) identified severe to extreme drought spells during 1986, 1988, 1998, 2000–2002, 2004, 2006, 2010, 2018–2019, and 2021 in most meteorological stations (met-stations). The Lasbella met-station experienced the most frequent extreme to severe droughts according to both the 12-month RDI (8.82%) and SPEI (15.38%) indices. The Dalbandin met-station (8.34%) follows closely behind for RDI, while Khuzdar (5.88%) comes in second for the 12-month SPEI. VCI data showed that Baluchistan experienced severe to extreme drought in 2000, 2001, 2006, and 2010. The most severe drought occurred in 2000 and 2001, affecting 69% of the study region. A positive correlation was indicated between meteorological (RDI, SPEI) and agricultural drought index (VCI). The multivariate indices can provide valuable knowledge about drought episodes and preparedness to mitigate drought impacts.

## Introduction

Climate change is causing variations in precipitation, evapotranspiration, global average temperatures drought occurrence and intensity. Human activities, like burning fossil fuels are altering weather patterns [1]. The climate change is increasing the frequency and intensity of

**Data Availability Statement:** The authors received the Rainfall and Temperature data from Pakistan Meteorological Department upon request. The authors did not receive any special privileges in

accessing the data. Individual researchers can contact them through the following Email and Contact numbers to obtain the required Data from Pakistan Meteorological Department. Address: Climate Data Processing Centre (CDPC), Karachi Phone: +92-21-99261413; +92-21-99261438 Email: info.cdpc@pmd.gov.pk Website: https://cdpc.pmd.gov.pk/

**Funding:** This Study was Supported by National Natural Science Foundation of Chin, 41877433 and Natural Science Foundation of Hebei Province, 18963301D. This research was also partly supported by Korea Environmental Industry & Technology Institute (KEITI) through Water Management Program for Drought Project, funded by Korea Ministry of Environment (MOE) (2022003610003).

**Competing interests:** The authors have declared that no competing interests exist.

extreme weather events worldwide [2,3]. No area on the earth is protected from the negative consequences of climate change [4], which are impacting the global water cycle by causing more extreme weather conditions, like droughts and floods [5]. Drought is a major natural hazard characterized by prolonged precipitation deficits, with significant impacts on agriculture, water availability for human well-being [6–10]. World population is expected to increase by about 25–30% in the coming three decades, the severity of droughts will pose a growing threat to food security in many countries [11]. According to the Emergency Events Database (EM-DAT), droughts have affected more than 200 million people worldwide from 1900 to 2016, resulting in the deaths of over 11 million people [12]. Droughts can force people to migrate, disrupting social networks, damaging crops and livestock, and reducing water availability for water-dependent activities [13,14]. Droughts are among the most costly and catastrophic natural disasters [15]. Therefore, drought early warning systems, along with identification and assessment mechanisms, are essential for agricultural regions to safeguard food production and prevent migration.

Multivariate drought indices are used to assess and monitor meteorological, agricultural, and hydrological drought events, as well as to evaluate their effects [16]. These indices helps to understand and characterize the meteorological, hydrological, socioeconomic, and agricultural droughts [17]. Meteorological drought leads to hydrological drought, characterized by reduced streamflow, lowered groundwater levels, and depleted soil moisture [18,19]. Agricultural drought occurs when crops fails due to insufficient water [20]. This happens when scarce rainfall, along with a decline in groundwater and surface water levels, causes a decrease in crop yields or leads total crop failures [21,22]. Similarly, socioeconomic drought occurs when drought affects the social structure and economy of a society [23]. Previous research studies have concluded that the use of the RDI is more effective for assessing agricultural drought [24,25]. The SPEI, a standardized drought index commonly used for assessing meteorological and agricultural drought at different timescales, has been found to be more effective for assessing drought impacts on crops at various growth stages [26–28]. Meanwhile, the VCI uses remote sensing data and is more useful index for monitoring agricultural drought.

Agriculture in Pakistan mostly relies on rainfall, and since most parts of the country experience a dry climate, except for the northern area which has humid conditions, the country is vulnerable to droughts [29–31]. Droughts occur in Pakistan four times out of every ten years [32]. Baluchistan is one of the regions most vulnerable to drought hazards due to its arid climate [33,34], where 85% of the population relies on agriculture for their livelihood [35]. Droughts in 1996, 2001, 2002, 2004, 2009, and 2014 had drastic impacts on the livelihood and economy of the province [34]. Similarly, severe drought conditions occurred in different parts of Balochistan, with Dalbandin and Quetta experiencing drought in 1999–2000; Barkhan, Dalbandin, Lasbella, and Sibi in 2002 and 2003; Zhob in 2010–2011; Kalat and Khuzdar in 2014 and 2015; and Panjgur in 2017–2018 [28]. The prolonged drought of 1998–2002 affected 80% of fruit orchards and was responsible for the death of two million animals in Baluchistan [36,37]. This drought also severely affected agricultural yield in Baluchistan and Sindh, with major crops such as wheat, cotton, and rice experienced a negative growth of almost 10%, while overall agricultural growth declined by 2.6% [38,39]. Nearly half of the apple, peach, and apricot orchards in upland Baluchistan were destroyed, with the trees being cut down and sold as firewood [40]. The drought of 1998–2002 affected more than three million people and two million livestock [41]. Frequent droughts, along with high water withdrawal for irrigation, have led to a decline in the water table in Baluchistan [40].

The novelty of this study lies in the utilization of multiple drought indices, including the RDI, SPEI, and VCI, along with run theory, to determine the spatiotemporal patterns, intensity, variability, and frequency of meteorological and agricultural drought in Baluchistan

province. The correlation coefficient statistical tool was applied to quantify the correlation between the meteorological and agricultural drought indices. These indices will serve as a reliable source of information on meteorological and agricultural drought, assisting in the formulation of appropriate adaptation strategies for the future.

## The study area

Baluchistan, the largest province in Pakistan, stretches from 24.89° to 32.098° North latitude and 60.87° to 70.30° East longitude (Fig 1). The province's climate varies from arid to hyper-arid, with precipitation mainly occurring from two sources: monsoon rains in the summer and winter precipitation in the eastern and northeastern parts brought by western depressions. Annual precipitation ranges from 30 mm to 397 mm [42,43]. The province's topography exhibits significant variation, ranging from high mountains to flat plains. Despite its vast area, Baluchistan is the least populated province, with approximately 8 million people, covering about 44% of Pakistan's total area [44]. Consequently, large areas of land remain sparsely

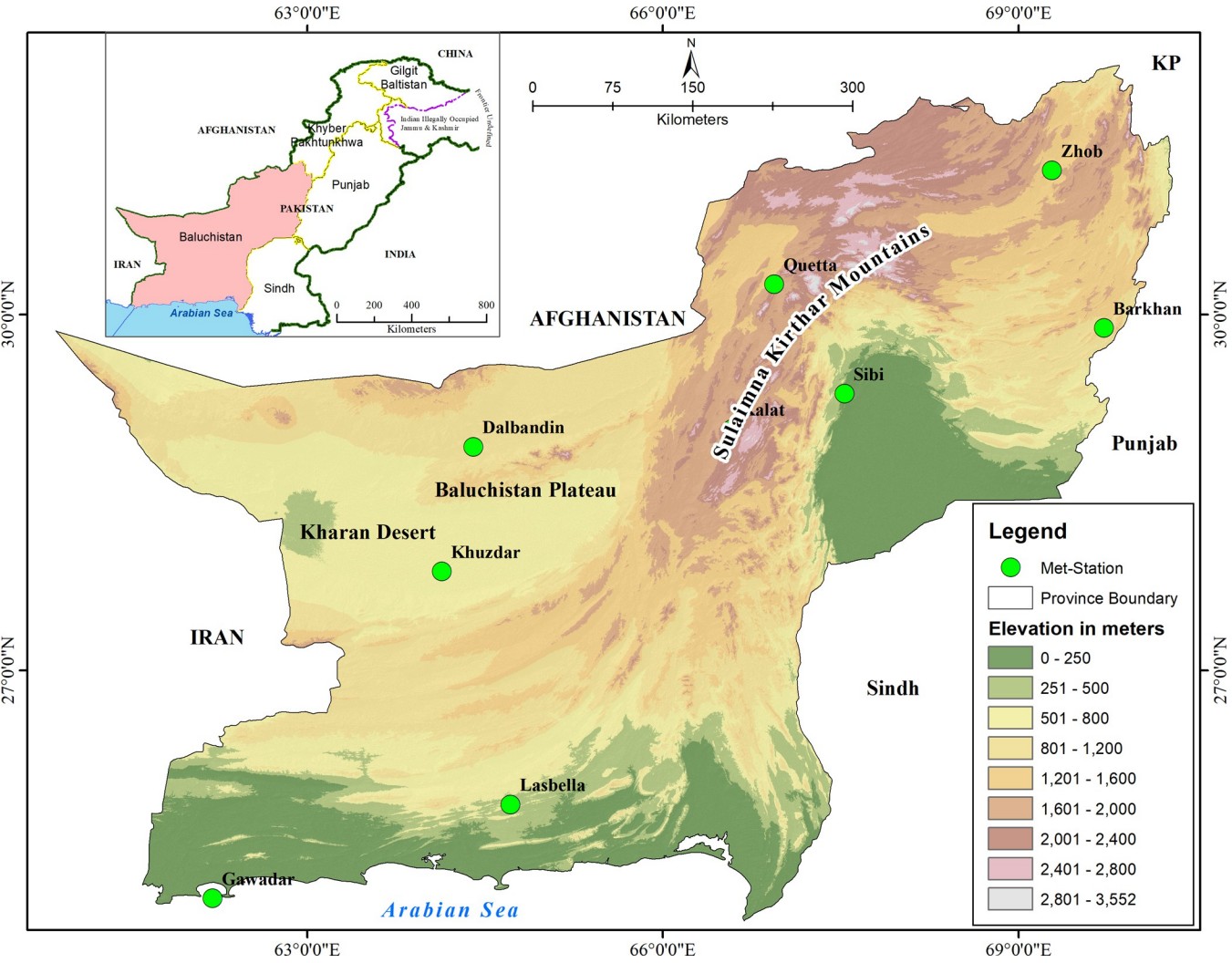

**Fig 1. Location and topography of the study area.**

populated. The province's strategic location, coupled with the development of the Gwadar port, positions it as a future gateway and hub for regional and global trade [45].

## Methods and material

### Data collection and method

To achieve the research objectives, mean temperature and precipitation data were collected for multiple meteorological stations from the Pakistan Meteorological Department (Table 1). Due to variations in the establishments dates of meteorological stations, the data period varied. Specifically, Dalbandin, Quetta, Khuzdar, and Sibi stations provided 36 years of data (1986–2021), while Lasbela and Kalat had 34 years (1988–2021), Zhob had 32 years (1990–2021), and Barkhan had 13 years (2009–2021) (Table 1). Inverse Distance Weight interpolation was employed to quantify precipitation distribution across the study area. The collected temperature and precipitation data were then used to calculate Potential Evapotranspiration (PET), essential for estimating the RDI and SPEI. The Thornthwaite method was employed for PET calculation in R. Additionally, R Studio was utilized to analyze SPEI and RDI results to identify drought trends and quantify drought severity using Run Theory techniques. The VCI for the growing season (April to July) from 1998 to 2021 was derived from the Normalized Difference Vegetation Index (NDVI), calculated from Landsat images obtained via Google Earth Engine. Drought assessment relied on RDI, SPEI, and VCI, with December's data used to prepare figures alongside the 12-month analysis of RDI and SPEI. Linear regression and correlation coefficient analyses were employed to quantify the relationship between the severity levels (ranging from non-drought to extreme drought) of these indices.

### Reconnaissance Drought Index (RDI)

The RDI, first proposed by [46] and later refined by [25], is commonly used for assessing meteorological and agricultural drought severity. Numerous studies have employed the RDI in various semiarid and arid regions, and its usage is expanding due to its high sensitivity and resilience, coupled with low data requirements [47–49]. This drought index is based on potential evapotranspiration (PET) and precipitation. The RDI can be expressed in three alternative mathematical notations: the initial value of the RDI ($\alpha k$), the normalized RDI (RDIn), and the standardized RDI (RDIst). When using a monthly time step and the aggregate form, the RDI

**Table 1. Locations, time periods and elevations of meteorological stations.**

| Met-station | Time Period | Latitude | Longitude | Elevation (m) |
|---|---|---|---|---|
| Barkhan | 2009–2021 | 29˚53'50.07"N | 69˚31'39.41"E | 1098 |
| Dalbandin | 1986–2021 | 28˚53'7.45"N | 64˚23'46.92"E | 850 |
| Kalat | 1988–2021 | 29˚1'49.3"N | 66˚35'16.36"E | 2017 |
| Lasbella | 1988–2021 | 25˚45'0" N | 66˚34'60" E | 54 |
| Quetta | 1986–2021 | 30˚10'47.42"N | 66˚58'29.9"E | 1600 |
| Khuzdar | 1986–2021 | 27˚48'43.27"N | 66˚36'42.09"E | 1232 |
| Sibi | 1986–2021 | 29˚32'22.39"N | 67˚52'33.27"E | 436 |
| Zhob | 1990–2021 | 31˚20'28.54"N | 69˚26'55.17"E | 1561 |

(αk) for the (ith) year is determined as the coefficient and is represented as follows:

$$a_k^{(i)} = \frac{\sum_j^i = Pij}{\sum_j^i = PETij}, i = 1 \text{ Not } N \tag{1}$$

With the following expression, the RDI$_n$ was calculated as.

$$RDI_{n(k)^{(i)}} = \frac{a_k^{(i)}}{\bar{a}k} - 1 \tag{2}$$

The values of RDI*st* are computed as.

$$RDI_{st^i}(k) = \frac{y_k^{(i)} - \bar{y}_k}{\hat{\sigma}_{yk}} \tag{3}$$

Where $y_k$ is the arithmetic mean of $\bar{y}_k$ and an is its standard deviation, $y_k$ is the lognormal of the values of $a_k$.

## Standardized Precipitation Evapotranspiration Index (SPEI)

The SPEI calculates the water balance between precipitation and PET to assess drought and wet conditions in a particular area. The SPEI considers the effect of temperature and precipitation on drought, and therefore combines the advantages of both the Standardized Precipitation Index (SPI) and the Palmer Drought Severity Index (PDSI) [50]. Most meteorological stations throughout the world do not directly calculate PET, so numerous techniques have been developed to calculate it indirectly from a few accessible meteorological factors [51]. Some famous methodologies to estimate potential evapotranspiration (PET) indirectly from a few accessible meteorological factors are the Penman-Monteith, Hargreaves and Thornthwaite methods [52,53]. In the present study, the Thornthwaite technique was applied to compute PET, and Table 2 shows the dry and wet conditions for different SPEI values.

The PET is represented by the Eq 4.

$$\bar{a}k^{PET=16\times} \left(\frac{N}{12}\right) \times \left(\frac{m}{30}\right) \times \left(10 \times \frac{T_i}{I}\right)^{\alpha} \tag{4}$$

The average monthly sunlight hours are denoted by N, the number of days in a month by m, the monthly average temperature in Celsius by Ti, and the coefficient dependent on I by α calculated based on Eq 5.

$$\alpha = 6.75 \times 10^{-7} \times I^3 - 7.71 \times 10^{-5} \times I^2 + 1.79 \times 10^{-2} \times I + 0.49 \tag{5}$$

**Table 2. Drought intensity classifications based on RDI and SPEI values.**

| RDI Value | Category |
|:---:|:---:|
| ≥ 2 | Extreme Wet |
| 1.5 to 1.99 | Severe Wet |
| 1.0 to 1.49 | Moderate Wet |
| − 0.99 to 0.99 | Near Normal |
| −1.0 to − 1.49 | Moderately Drought Condition |
| − 1.5 to − 1.99 | Severely Drought Condition |
| ≤ −2.0 | Extremely Drought Condition |

The following expression expresses the thermal index, which is obtained from the total of the 12-month index values (Eq 6).

$$I = \sum_{i=1}^{12} X\left(\frac{T_i}{5}\right)^{1.514} \tag{6}$$

The difference between precipitation and PET (water balance) was calculated using the Eq 7.

$$Di = Pi - PETi \tag{7}$$

The Di is the water balance calculated from the difference between precipitation and PET. It shows whether there is a water surplus or deficit for the month under consideration. The Di results are accumulated over various time frames using the Eq 8.

$$D_n^k = \sum_{i-1}^{k-1}(pn - 1 - PETn - 1), n \geq k \tag{8}$$

The time scale of the data is represented by the variable k, and the computation frequency is represented by the variable n in this equation. SPEI differs from SPI in that it requires three distribution parameters: Pearson III, lognormal, and extreme values, while SPI only requires two. In a two-parameter distribution, the variable x has a lower value limit of zero ($0 > x1$), but in a three-parameter distribution, x can take on values in the range ($>x1$), where x1 is the distribution's origin parameter. The variable x can take on negative values, and negative values are common in the D series. The log-logistic probability density function was used to represent the Di values using the three parameters.

$$f(x) = \frac{\beta}{\alpha}\left(\frac{x-\gamma}{\alpha}\right)\beta - 1\left[1 + \frac{x-\gamma}{\alpha}\right]^{-2} \tag{9}$$

The L-moment method plays a central role in determining the three crucial parameters: $\alpha$, the shape parameter $\beta$, and the origin parameter $\gamma$. These parameters are obtained through the equations illustrated below:

$$\alpha = \frac{(wo - 2w1)\beta}{\Gamma\left(1 + \frac{1}{\beta}\right)\Gamma\left(1 - \frac{1}{\beta}\right)} \tag{10}$$

$$\beta = \frac{(2w1 - w0)}{6w1 - w0 - 6w2} \tag{11}$$

$$\gamma = w0 - \alpha\Gamma\left(1 + \frac{1}{\beta}\right)\Gamma\left(1 - \frac{1}{\beta}\right) \tag{12}$$

The expression captures the probability density function of the log-logistic distribution employed to model the D series data, where the term $\Gamma(\beta)$ represents the gamma function applied to $\beta$:

$$F(X) = \left[\left(1 + \frac{\alpha}{x-\gamma}\right)^{\beta}\right] - 1 \tag{13}$$

The Eq (13) below can determine the SPEI directly from the standardized values of F(x):

$$\text{SPEI} = \text{W} - \frac{\text{C0} + \text{C1W} + \text{C2Wd} + \text{bc}}{1 + \text{d1w} + \text{d2w2} + \text{d3w3}} \tag{14}$$

Where $W = \sqrt{-2ln(p)}$, for P ≤ 0.5 and P is the probability of exceeding a determined D Value with P = 1- F (X); when P > 0.5 in the above equations the given below values are constant

C0=2.515515   C1=0.802853   C2=0.010328   d1 = 1.432788
d2 = 0.189269   d3 = 0.001308

The SPEI zero signifies normal precipitation conditions. Positive numbers signify wetter-than-usual periods, while negative values reveal drought conditions (Table 2).

## Vegetation Condition Index (VCI)

The severity of agricultural drought during the growing season was assessed using the VCI, which is calculated from NDVI [54]. The Vegetation Condition Index is determined by calculating the long-term minimum NDVI ($NDVI_{mini}$) and long-term maximum NDVI ($NDVI_{max}$) over the same time period. These values are then compared to the NDVI of the current period ($NDVI_j$).

$$\text{VCI} = \left( \frac{NDVI_j - NDVI_{mini}}{NDVI_{max} - NDVI_{mini}} \right) \times 100 \tag{15}$$

We utilized Google Earth Engine to acquire mean annual NDVI data for our study area. To ensure data quality and minimize potential disruptions in the images, we applied a filter to exclude imagery with greater than 10% cloud cover. It is noteworthy that the study area, characterized as an arid to semi-arid region, experiences minimal annual cloud cover. Subsequently, the acquired data was processed in ArcGIS to calculate the VCI.

The VCI is calculated to assess drought impacts on vegetation. VCI is especially helpful for agriculture because it evaluates variations in NDVI over time. The Vegetation Condition Index for the growing season (April to July) was calculated from 1998 to 2021, covering a period of 24 years. During April to July, maximum area of the province is covered with crops and natural vegetation, therefore we selected this period for the VCI analysis. VCI values are expressed as percentages, ranging from 0 (lowest) to 100 (highest) (Table 3). Values equal to or below 50% are considered to indicate various severities of drought.

## Results

### Spatio-temporal variation in mean annual and mean monthly precipitation

Precipitation in the study area mainly occurs during the winter and summer seasons, with varying rates across different regions of Baluchistan. Regions receiving less than 250 mm of

**Table 3. Drought severity classifications based on VCI values.**

| Drought Severity | VCI |
|---|---|
| Extreme | 0 < VCI < 10 |
| Severe | 10 <VCI < 20 |
| Moderate | 20 <VCI < 35 |
| Mild | 35 <VCI < 50 |
| No drought | 50 < VCI < 100 |

precipitation are classified as arid, those with precipitation ranging from 250–750 mm are classified as semi-arid, while areas receiving over 750 mm annually are termed humid [55,56]. Based on this classification, Barkhan (420 mm) and Khuzdar (253.8 mm) are categorized as semi-arid regions. The highest annual rainfall was observed in Kalat at 982.3 mm (Table 4). Statistical analysis was used to quantify the central tendency and dispersion in precipitation data. The variations between maximum, minimum, and standard deviation in precipitation at different met-stations are shown in Table 4. The highest standard deviation is found in Kalat (163.8mm), followed by Barkhan (158.4 mm), indicating the highest variation in mean annual precipitation. Similarly, the highest standard deviation in mean monthly precipitation is recorded in Barkhan (30.9 mm). Normality of the precipitation data was tested using skewness and kurtosis. Skewness measures the symmetry and precision of the data. The annual precipitation data shows the highest positive skewness (right skewed) observed in Kalat with a skewness value of 3.3, and Barkhan with a value of 1.6 (Table 4). Similarly, the highest positive value of kurtosis was noted at Kalat (13.5) and Barkhan (2.9) meteorological stations. The highest negative kurtosis was observed at Dalbandin (-0.7), with Zhob meteorological station recording a kurtosis of -0.4 (Table 4).

The months of June, July, and August typically receive the highest amount of precipitation in these regions, while other meteorological stations, including Quetta, are classified as arid. Elevated areas such as Dalbandin, Kalat, and Quetta receive most of their rainfall during the winter months (December, January, and February) (Fig 2). In winter, due to the Siberian wind, the temperature becomes low in Dalbandin, Kalat, and Quetta, which receive maximum precipitation. Kalat recorded the highest precipitation in winter (119.9 mm), while Khuzdar observed the lowest (18.6 mm). In contrast, Barkhan, Khuzdar, Lasbella, Sibi, and Zhob receive more precipitation during the summer months (June, July, and August) due to the monsoon (Table 5). The summer months collectively receive 679.8 mm of mean monthly rainfall, with Barkhan (237.6 mm) and Zhob (118.7 mm) recording the highest precipitation.

## Temporal variation of drought based on 1, 3 and 6-months RDI

The current study examines the spatial and temporal aspects of the 1, 3, and 6-month RDI. Severe to extreme drought conditions were observed in the 1, 3, and 6-month RDI values for the years 1986, 1988, 1991, 1998–2000, 2002, 2004, 2006, 2010, 2014, 2016, and 2018, which were common drought years across most weather stations in the study area. Fig 3 illustrates four distinct dry spells between 1986 and 2021, as indicated by the total RDI values for 1, 3, and 6-month periods.

**Table 4. Precipitation distribution by meteorological station.**

| Met-station | Min | Max | Mean | SD | Skewness | Kurtosis |
|---|---|---|---|---|---|---|
| Barkhan | 252 | 829.5 | 420 | 158.4 | 1.6 | 2.9 |
| Dalbandin | 3.5 | 182 | 81.9 | 48 | 0.3 | -0.7 |
| Kalat | 47 | 982.3 | 218.1 | 163.8 | 3.3 | 13.5 |
| Khuzdar | 89.6 | 594.7 | 253.8 | 125.8 | 1.2 | 1.1 |
| Lasbella | 8.7 | 474.6 | 191.1 | 110 | 0.7 | 0.3 |
| Quetta | 31 | 459.0 | 236.2 | 102.9 | 0.3 | -0.4 |
| Sibi | 9.7 | 471.9 | 204 | 101.2 | 0.4 | -0.02 |
| Zhob | 109.6 | 495 | 284.5 | 102.8 | 0.4 | -0.4 |

**Note:** The unit of precipitation is millimeter.

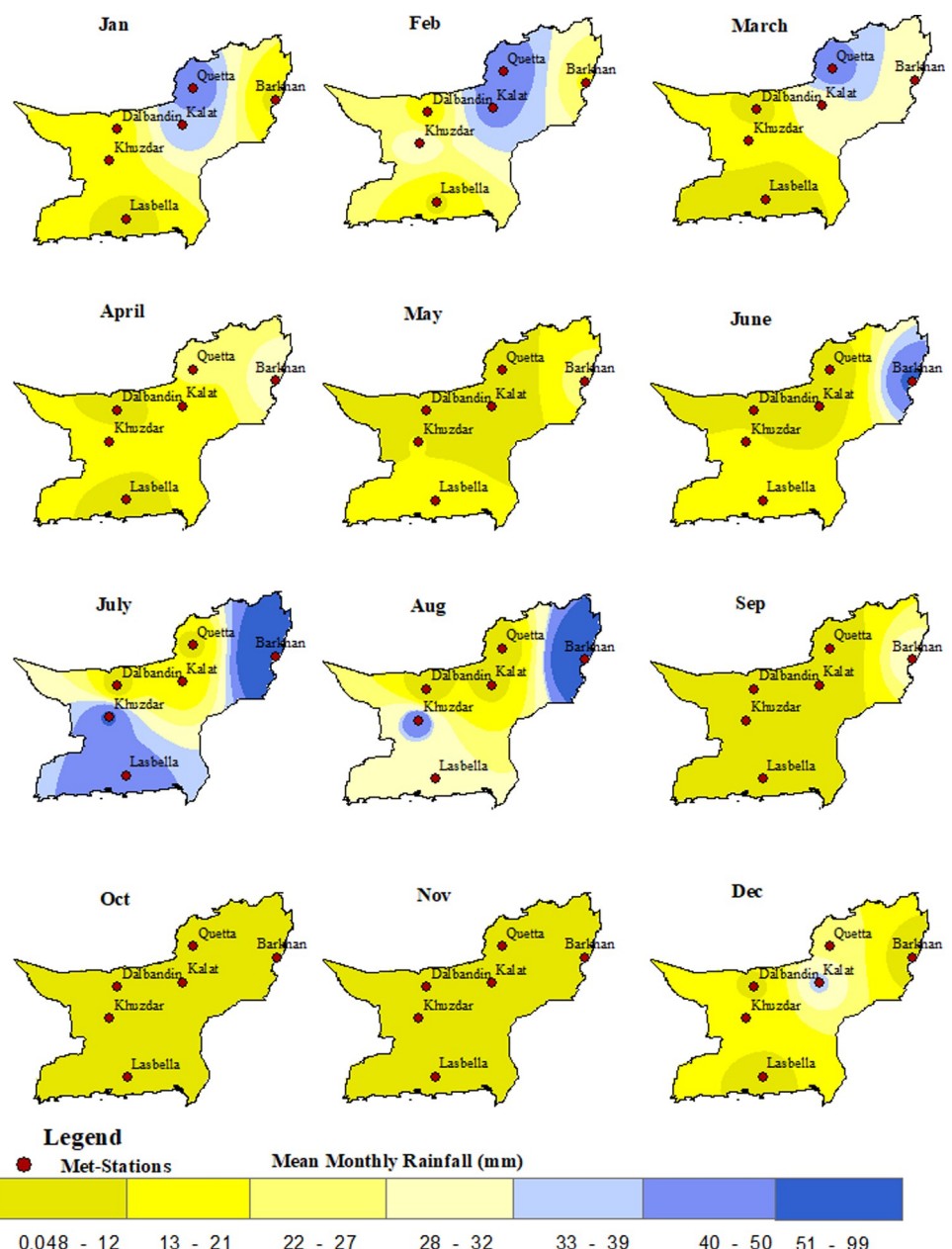

**Fig 2. Distribution of mean monthly precipitation in Baluchistan province.**

The RDI analysis reveals distinct phases of drought occurrence over the study period. The first dry phase occurred in the early 1990s, followed by increased rainfall in the latter part of the decade. Subsequently, during 2000–2002, most meteorological stations exhibited elevated 1, 3, and 6-month RDI values, indicating the onset of a second drought phase. This was followed by a wet season with above-average precipitation. In 2010, the severity of drought was alleviated across most met-stations, as indicated by the 1, 3, and 6-month RDI values. The severe to extreme drought observed during 2016–2018 represents the fourth phase of the dry spell. A majority of meteorological stations showed trends of severe to extreme RDI values (1, 3, and 6 Months) during the last two decades (2000–2020). Sibi and Zhob met-stations

**Table 5. Distribution of mean monthly precipitation across the study area.**

| Met-stations | Winter | | | Spring | | | Summer | | | Autumn | | | |
|---|---|---|---|---|---|---|---|---|---|---|---|---|---|
| | Dec | Jan | Feb | Mar | Apr | May | Jun | Jul | Aug | Sep | Oct | Nov | SD |
| Barkhan | 6.1 | 10.3 | 20.4 | 30.7 | 32.0 | 25.9 | 52.3 | 99.5 | 85.9 | 33.1 | 9.7 | 4.0 | 30.9 |
| Dalbandin | 9.0 | 16.2 | 15.6 | 18.7 | 4.7 | 1.0 | 3.6 | 3.4 | 0.3 | 0.2 | 1.4 | 3.0 | 7.0 |
| Kalat | 34.1 | 37.2 | 48.5 | 34.9 | 11.6 | 4.2 | 8.6 | 19.8 | 9.1 | 5.9 | 3.9 | 5.9 | 15.7 |
| Khuzdar | 9.9 | 5.1 | 3.7 | 14.5 | 16.5 | 29.8 | 26.9 | 16.5 | 14.4 | 15.6 | 56.5 | 50.4 | 17.1 |
| Lasbella | 8.2 | 6.2 | 11.0 | 11.5 | 9.2 | 20.5 | 16.6 | 54.7 | 32.5 | 8.0 | 6.3 | 1.5 | 15.3 |
| Quetta | 27.3 | 45.8 | 46.3 | 46.0 | 23.6 | 6.9 | 4.0 | 10.3 | 4.2 | 4.1 | 4.1 | 8.6 | 18.5 |
| Sibi | 5.6 | 10.7 | 19.6 | 22.6 | 8.8 | 6.9 | 18.2 | 39.5 | 40.9 | 15.2 | 2.8 | 1.3 | 13.3 |
| Zhob | 8.7 | 13.8 | 27.0 | 35.7 | 28.7 | 14.8 | 23.4 | 57.7 | 37.6 | 12.1 | 4.1 | 7.6 | 15.8 |

exhibited a notable number of severe to extreme drought trends in the 1, 3, and 6-month RDI values (Fig 3).

The frequencies of the 1, 3, and 6-month RDI were determined using run theory in R Studio (Table 6). The results revealed maximum extreme drought frequencies in Barkhan (1.9%, 2.6%, and 1.3%), followed by Khuzdar (0.0%, 2.1%, and 2.1%), and Quetta (0.0%, 1.5%, and 4.2%) meteorological stations. Similarly, the highest frequencies of severe drought in the 1, 3, and 6-month RDI were observed in Quetta (3.2%, 5.4%, and 2.5%), Zhob (2.3%, 4.0%, and 3.5%), and Kalat (0.2%, 3.0%, and 6.0%) meteorological stations. The frequency of drought events at Barkhan meteorological station was notably high, which could be attributed to limited meteorological data availability. Additionally, the frequencies of moderate drought varied among stations, but the stations most affected by moderate drought frequencies in the 1, 3, and 6-month RDI were Zhob (8.8%, 8.1%, and 9.6%) and Quetta (4.2%, 8.9%, and 7.5%) meteorological stations.

## Drought characteristics based on 12-month RDI

The spatial and temporal characteristics, severity, and frequencies of drought were assessed using the 12-month RDI across various parts of Baluchistan province (Fig 3). The results indicated that the majority of the study area experienced severe to extreme drought conditions. Extreme drought spells were particularly notable in hyper-arid regions such as Dalbandin (2000, 2002) and semi-arid areas like Lasbella (2002) and Sibi (2002). Additionally, severe drought conditions were observed across all meteorological stations. Analysis of the 12-month RDI revealed several severe dry spells occurring between 1986 and 2021. The most common severe drought events were observed across the majority of stations during the years 2000–2002, 2004, 2010, and 2018. Khuzdar (1991, 2002, 2004) emerged as one of the most affected stations by severe drought, followed by Zhob (2001, 2011, 2016), Lasbella (2000, 2004), and Quetta (2001, 2018). Short-term moderate drought spells were also observed in some meteorological stations within the study area in 1998, 2000, 2001, 2004, 2006, 2014, and 2018 (Fig 3). Furthermore, a longer-lasting moderate drought occurred across the majority of meteorological stations in the study area from 2000 to 2001. The results of the 12-month RDI data (Fig 3) revealed fluctuations in drought trends across most meteorological stations, particularly between 2010 and 2015. Moreover, the stations Lasbella and Sibi exhibited a more consistent tendency towards severe to extreme drought compared to other stations.

The analysis of the 12-month RDI highlighted that Dalbandin exhibited the highest frequency of extreme drought events, at 5.56%, followed by Lasbella (2.94%) and Sibi (2.78%) (Table 7). Similarly, Khuzdar and Zhob recorded the highest frequency of severe drought, at

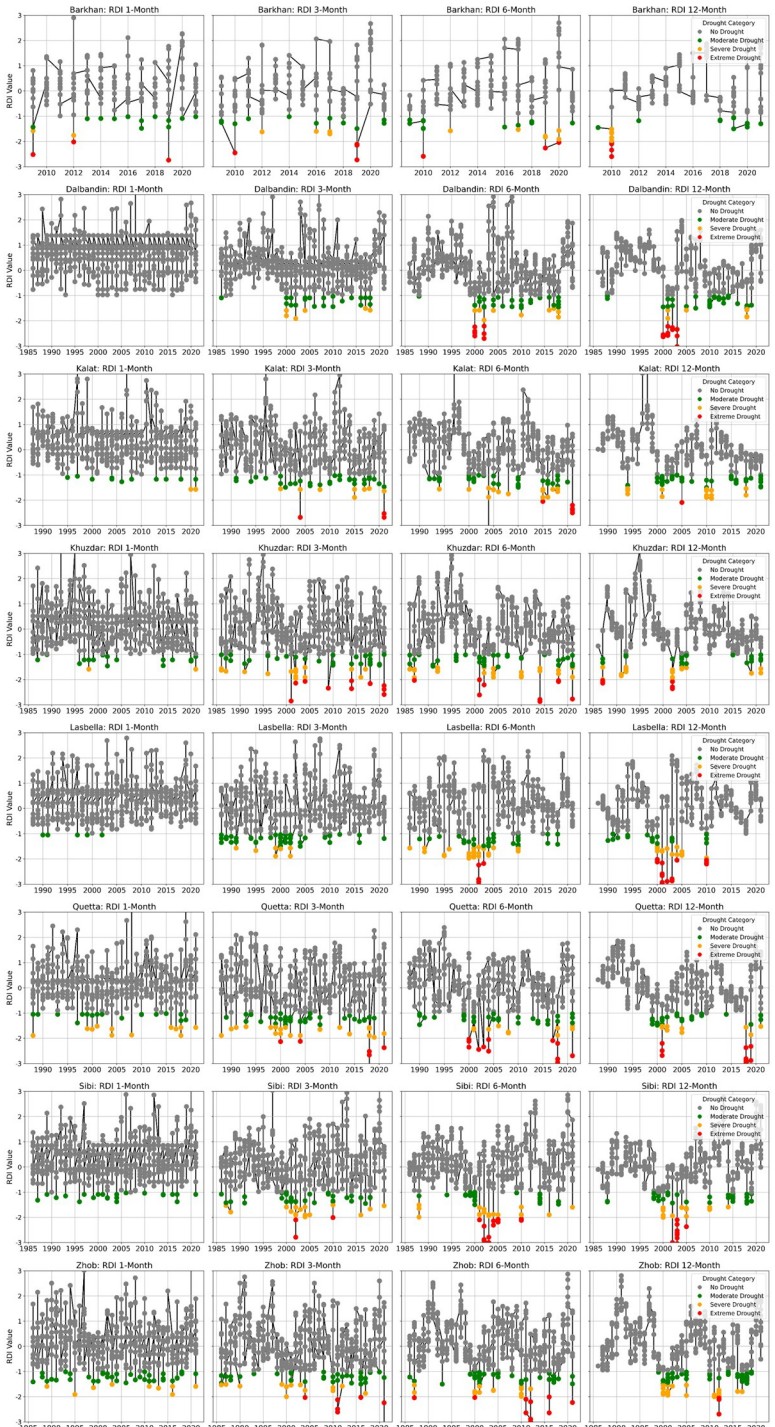

**Fig 3. Temporal distribution of drought trends in 1, 3, 6, and 12-Month RDI in Baluchistan.**

8.33%, followed by Barkhan (7.69%), and Lasbella and Quetta, each with a severe drought frequency of 5.88%. Among the meteorological stations, Barkhan registered the highest frequency of moderate drought (15.38%), followed by Kalat (14.71%), Quetta (8.82%), and Sibi (8.33%). The analysis of the 12-month RDI unveiled severe to extreme drought occurrences in

**Table 6. Drought frequency based on severity categories of the 1, 3 and 6-month RDI.**

| Met-Station | 1-Month RDI | | | 3-Month RDI | | | 6-Month RDI | | |
|---|---|---|---|---|---|---|---|---|---|
| | Moderate Drought | Severe Drought | Extreme Drought | Moderate Drought | Severe Drought | Extreme Drought | Moderate Drought | Severe Drought | Extreme Drought |
| | (-1.0 to -1.49) | (-1.5 to -1.99) | (< -2.0) | (-1.0 to -1.49) | (-1.5 to -1.99) | (< -2.0) | (-1.0 to -1.49) | (-1.5 to -1.99) | (< -2.0) |
| Barkhan | 6.4 | 1.9 | 1.9 | 7.9 | 4.6 | 2.6 | 7.3 | 2.0 | 1.3 |
| Dalbandin | 0.0 | 0.0 | 0.0 | 5.6 | 1.4 | 0.0 | 7.5 | 2.1 | 1.6 |
| Kalat | 3.4 | 0.2 | 0.0 | 8.1 | 3.0 | 0.2 | 10.2 | 6.0 | 0.5 |
| Khuzdar | 3.7 | 0.5 | 0.0 | 7.9 | 3.7 | 2.1 | 8.2 | 5.4 | 2.1 |
| Lasbella | 1.2 | 0.0 | 0.0 | 11.4 | 2.2 | 0.0 | 7.9 | 6.9 | 1.5 |
| Quetta | 4.2 | 3.2 | 0.0 | 8.9 | 5.4 | 1.5 | 7.5 | 2.5 | 4.2 |
| Sibi | 5.8 | 0.0 | 0.0 | 8.2 | 4.2 | 0.5 | 6.8 | 4.2 | 3.3 |
| Zhob | 8.8 | 2.3 | 0.0 | 8.1 | 4.0 | 1.6 | 9.6 | 3.5 | 2.6 |

Baluchistan in recent years, particularly in the southwestern regions. Notably, the Dalbandin station emerged as the most significantly affected by extreme drought, followed by Kalat and Khuzdar.

## Temporal variation of drought based on 1, 3, and 6-months SPEI

The 1, 3, and 6-month SPEI were calculated for eight meteorological stations to assess drought conditions in the study area. These indices depict extreme, moderate, and wet conditions across the region. Among the stations, Kalat exhibited the highest susceptibility to severe to extreme drought events (occurring in 1993, 1995, 2000, 2005, 2011, 2018, 2019, and 2021), followed by Khuzdar (1991–1992, 2002, 2004, 2010, 2012, 2014, 2016, and 2018) and Quetta (1994, 1998, 2000, 2001, 2006, 2010, 2014, 2016, and 2018). Severe to extreme drought conditions in the 1, 3, and 6-month SPEI were observed across the majority of stations during the years 1986, 1988, 1991, 1993, 1998–2000, 2002, 2004, 2006, 2010, 2011, 2014, 2016, 2018, and 2021. Similar to the RDI, the results from SPEI also revealed distinct phases during the study period (1986–2021). The analysis of 1, 3, and 6-month SPEI identified the first phase of severe to extreme drought in 1993 across the majority of meteorological stations, followed by a subsequent wet season with above-normal precipitation. The second phase of severe to extreme drought occurred from 2000 to 2002 across most of the meteorological stations. The severe to extreme drought of 2010 marked the third phase of the dry season. The final phase occurred between 2016 and 2018 across most parts of the study area. Similar to the RDI, the majority of meteorological stations showed trends of severe to extreme drought in the 1, 3, and 6-month

**Table 7. Drought frequency across different severity categories of the 12-month RDI.**

| Met-stations | Moderate Drought (-1.0 to -1.49) | Severe Drought (-1.5 to -1.99) | Extreme drought (< -2.0) |
|---|---|---|---|
| Barkhan | 15.38 | 7.69 | 0.00 |
| Dalbandin | 5.56 | 2.78 | 5.56 |
| Kalat | 14.71 | 2.94 | 0.00 |
| Khuzdar | 5.56 | 8.33 | 0.00 |
| Lasbella | 5.88 | 5.88 | 2.94 |
| Quetta | 8.82 | 5.88 | 0.00 |
| Sibi | 8.33 | 2.78 | 2.78 |
| Zhob | 5.56 | 8.33 | 0.00 |

SPEI values during the last two decades from 2000 to 2020 (Fig 4). The meteorological stations exhibiting a number of severe to extreme drought trends included Sibi and Quetta in the 1, 3, and 6-month SPEI.

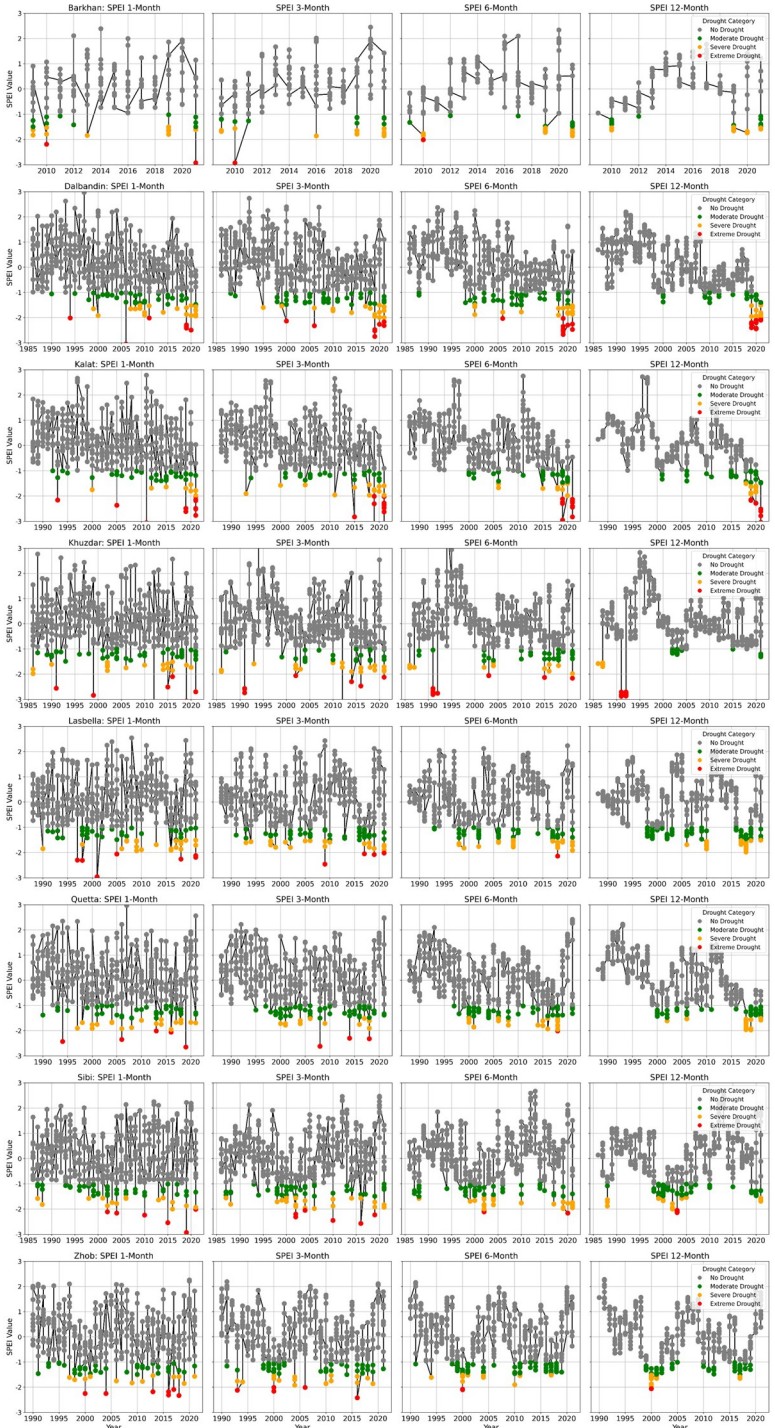

**Fig 4. Temporal distribution of drought trends in 1, 3, 6, and 12-Month SPEI in Baluchistan.**

Severe to extreme drought conditions were observed in Kalat, Khuzdar, Quetta, and Zhob in the 1, 3, and 6-month SPEI (Table 8). The results revealed that the maximum extreme drought frequencies were noted in Kalat (2.7%, 2.5%, and 3.5%), followed by Zhob (1.8%, 7.1%, and 0.5%) and Khuzdar (1.6%, 1.9%, and 2.3%). Similarly, the highest severe drought frequencies in the 1, 3, and 6-month SPEI were observed in Barkhan (8.3%, 8.5%, and 8.7%), Lasbella (5.4%, 5.7%, and 6.5%), Quetta (5.1%, 3.2%, and 6.5%), and Khuzdar (4.2%, 5.1%, and 3.5%). The frequencies of moderate drought varied from station to station, with Zhob (10.7%, 11.3%, and 16.5%) and Quetta (8.3%, 13.1%, and 11.4%) being the most affected stations in terms of moderate drought frequencies according to the 1, 3, and 6-month SPEI (Fig 4 and Table 8).

## Distribution of drought based on the 12-month SPEI

The 12-month SPEI was calculated to assess fluctuations in precipitation and PET in the study region. The results revealed severe to extreme drought spells during the research period (1986–2021). The Dalbandin station was highly affected by extreme drought in 2019 and 2021, followed by Kalat in 2021, and Khuzdar in 1991. In the 12-month SPEI, severe drought conditions were prevalent in some stations during 1998–1999, 2001–2002, 2004, 2006, 2010, and 2018. Barkhan experienced severe drought in 2019 and 2021, Lasbella in 2017 and 2021, and Quetta in 2018 and 2021. The 12-month SPEI indicated a trend towards moderate to extreme drought in most of the meteorological stations over the last decade (Fig 4). Notably, the stations in Lasbella and Sibi exhibited a stronger tendency towards severe to extreme drought conditions.

The drought frequency in Baluchistan was assessed using the 12-month SPEI to determine the rate of drought recurrence. The results revealed that the Dalbandin meteorological station had the highest extreme drought frequency, at 5.56%, followed by Kalat (2.94%) and both Khuzdar and Sibi (2.78%) (Table 9). Similarly, the highest frequencies of severe drought were noted in Barkhan (15.38%), with Lasbella and Quetta each recording a frequency of 5.88%. Lasbella exhibited the highest moderate drought frequency (15.38%), followed by Sibi (16.67%), Zhob (15.63%), and Quetta (11.76%) (Table 9). The 12-month SPEI results revealed that severe to extreme drought spells have occurred in Baluchistan in recent years, particularly in the southwestern parts.

**Table 8. Drought frequency based on severity categories of the 1, 3, and 6-month SPEI.**

| Met-Station | 1-Month SPEI | | | 3-Month SPEI | | | 6-Month SPEI | | |
|---|---|---|---|---|---|---|---|---|---|
| | Moderate Drought | Severe Drought | Extreme Drought | Moderate Drought | Severe Drought | Extreme Drought | Moderate Drought | Severe Drought | Extreme Drought |
| | (-1.0 to -1.49) | (-1.5 to -1.99) | ($<$ -2.0) | (-1.0 to -1.49) | (-1.5 to -1.99) | ($<$ -2.0) | (-1.0 to -1.49) | (-1.5 to -1.99) | ($<$ -2.0) |
| Barkhan | 5.8 | 8.3 | 1.3 | 6.5 | 8.5 | 0.7 | 5.3 | 8.7 | 0.7 |
| Dalbandin | 7.6 | 4.8 | 1.6 | 9.3 | 4.2 | 1.9 | 7.7 | 4.9 | 2.1 |
| Kalat | 7.6 | 1.7 | 2.7 | 5.9 | 3.2 | 2.5 | 7.7 | 2.2 | 3.5 |
| Khuzdar | 8.3 | 4.2 | 1.6 | 6.0 | 5.1 | 1.9 | 7.3 | 3.5 | 2.3 |
| Lasbella | 7.6 | 5.4 | 1.7 | 11.6 | 5.7 | 1.0 | 12.2 | 6.5 | 0.2 |
| Quetta | 8.3 | 5.1 | 1.2 | 13.1 | 3.2 | 0.7 | 11.4 | 6.0 | 0.2 |
| Sibi | 9.7 | 3.5 | 1.4 | 9.3 | 4.9 | 1.4 | 11.5 | 5.9 | 0.9 |
| Zhob | 10.7 | 3.6 | 1.8 | 11.3 | 1.3 | 7.1 | 16.5 | 3.1 | 0.5 |

**Table 9. Drought frequency by severity categories of the 12-month SPEI.**

| Met-stations | Moderate Drought (-1.0 to -1.49) | Severe Drought (-1.5 to -1.99) | Extreme drought ($<$ -2.0) |
|---|---|---|---|
| Barkhan | 0.00 | 15.38 | 0.00 |
| Dalbandin | 5.56 | 0.00 | 5.56 |
| Kalat | 8.82 | 2.94 | 2.94 |
| Khuzdar | 8.33 | 2.78 | 2.78 |
| Lasbella | 17.76 | 5.88 | 0.00 |
| Quetta | 11.76 | 5.88 | 0.00 |
| Sibi | 16.67 | 5.56 | 0.00 |
| Zhob | 15.63 | 3.13 | 0.00 |

## Distribution of drought based on vegetation condition index

Drought severely affects the water supply for both human and natural purposes. The intensities of agricultural drought were assessed for each growing season over 24 years using the VCI. The maps created (Fig 5) illustrate the fluctuations in drought intensity across time and space in Baluchistan. The VCI was derived from the NDVI based on Landsat images of the growing season (April to July) from 1998 to 2021. The results of the VCI indicated variations from extreme drought to normal conditions. Extreme to severe drought conditions were found in the growing seasons of 2000–2001, 2006, and 2010. Severe drought conditions affected the growing season to a lesser extent in 1998, 2003, 2004, 2007–2008, and 2012. The results mainly revealed that severe drought conditions affected the central and southeastern parts of the study area, while normal conditions were primarily observed in the northeastern parts of the province.

The spatial maps of the Vegetation Condition Index (VCI) (Fig 5) show that moderate to extreme drought conditions were more common from 2000 to 2010. A shift from moderate to extreme drought was observed from 2011 onwards, peaking again in 2018 in the province. The VCI results revealed that extreme drought in the growing season was most notable in 2001, covering the largest area at 51.1%. Based on the results, severe to extreme drought from 2000 to 2010 reached its highest percentage in 2000 and 2001, covering nearly 69% of the province's area (Fig 6). Following 2000 and 2001, severe to extreme droughts were noted in the years 2004 (53%), 2006 (56%), 2008 (22%), and 2010 (47%). The lowest levels of moderate to extreme drought were observed in 2020 (1.97%), followed by 2019 (4.37%), and 2021 (2.25%) of the study area. During this period (1998 to 2021), 2020 was the year with the maximum normal conditions, recorded at a higher percentage (96%). Overall, the percentage of drought has decreased in recent years in the study area. However, it is important to note that drought remains a major concern in the region, and measures must be implemented to mitigate its effects.

## Statistical analysis through Pearson correlation coefficient and linear regression

Correlation analysis was conducted to understand the relationship between the percentage of RDI-VCI and SPEI-VCI indices on a 12-month timescale. A scatter plot was used to calculate the correlation coefficients between the meteorological and vegetation drought indices over 23 years (1998–2021). The results showed a positive correlation between the two sets of indices (Fig 7). The calculated correlation coefficients indicated a positive correlation between the 12-month RDI and VCI. This implies that when RDI values for drought conditions (normal to extreme) are high, VCI values for drought conditions also tend to be high, and vice versa. A

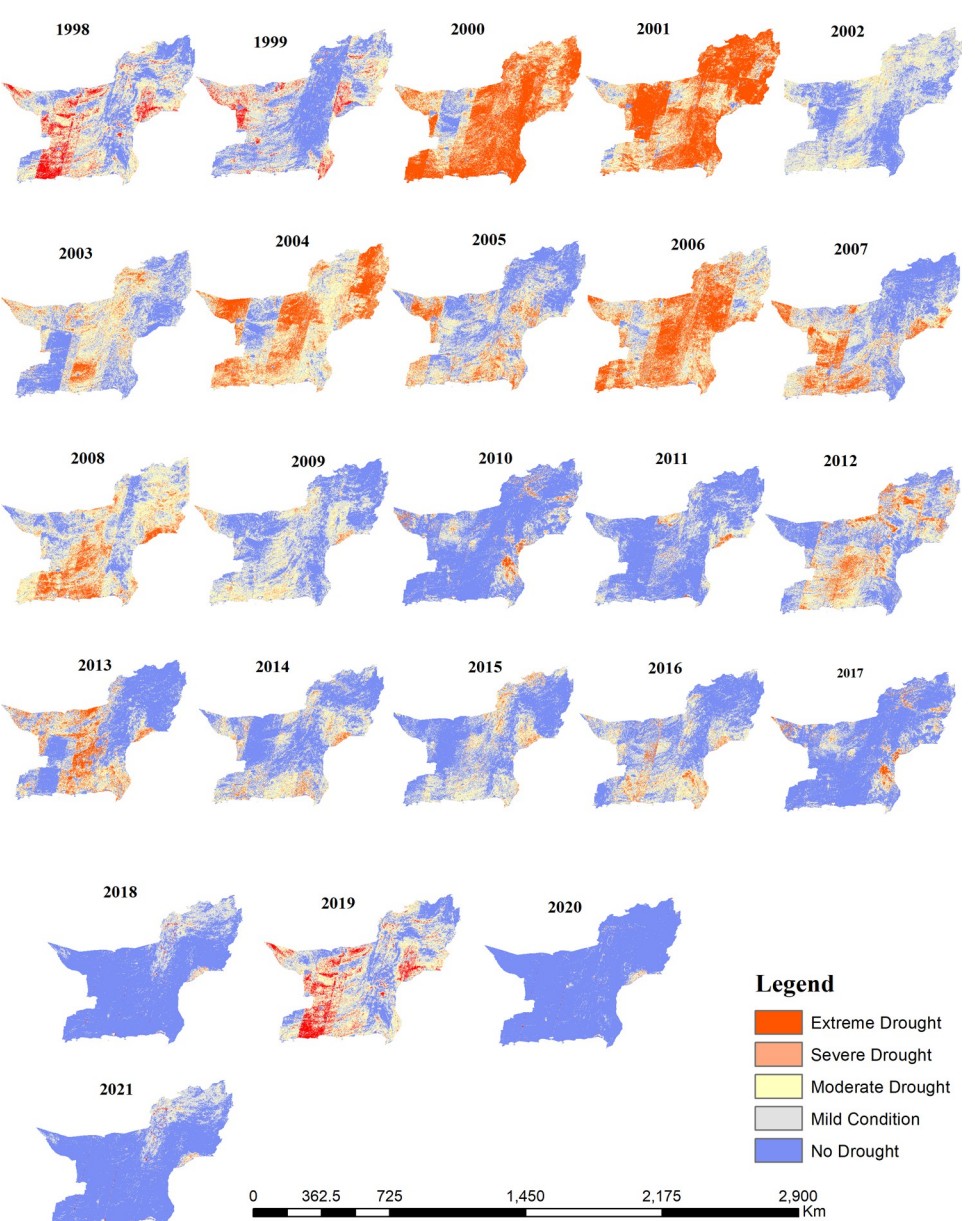

**Fig 5. Spatio-temporal distribution of drought using annual VCI in Baluchistan.**

positive correlation coefficient of 0.37 was observed between RDI and VCI values from 1998 to 2021. Additionally, the coefficient of determination (R2) exhibited high values between actual and estimated values. The R2 for RDI and VCI across the 23-year period was 0.21. The analysis revealed a positive correlation between varying drought conditions (normal to extreme drought) of the RDI and VCI.

Similarly, the RDI and VCI demonstrated a positive correlation, and the 12-month SPEI and VCI also exhibited positive correlation during the growing season (1998–2021). The calculated correlation coefficient was 0.27. Additionally, the coefficient of determination (R2) was 0.09, indicating a moderate relationship between the two indices. This suggests that SPEI and VCI can be used to assess drought conditions in the region (Fig 8).

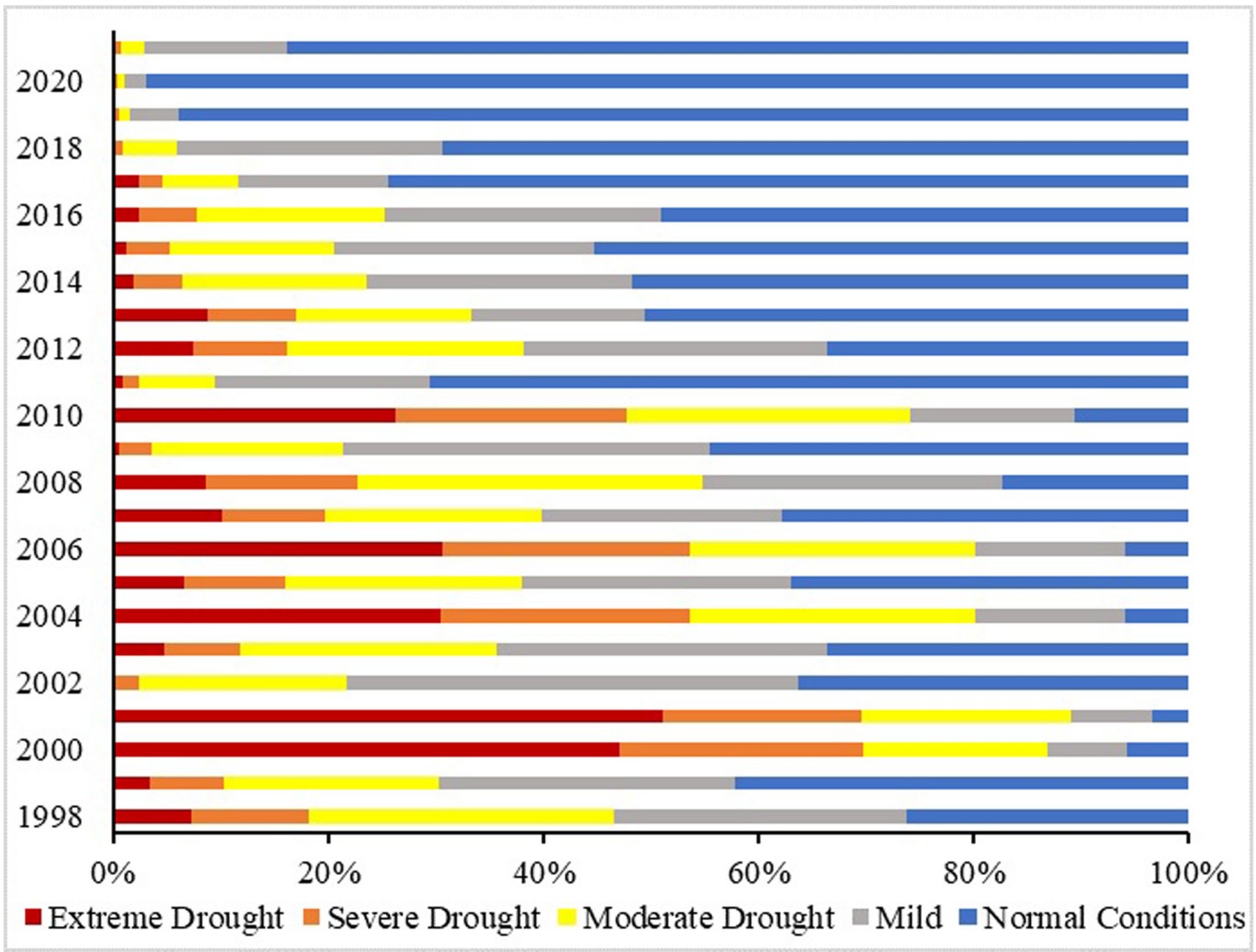

**Fig 6. Percentage of area affected by drought based on VCI.**

Table 10 provides insights into the relationship among the three drought indices i.e. SPEI, RDI and VCI based on Pearson correlation matrix. The results suggests a strong positive correlation between SPEI and RDI. The relationship among SPEI and VCI is found moderate positive that suggests that while SPEI has some influence on VCI but there might be other factors (soil moisture, agricultural practices, land use etc.) which also affect the vegetation health. The correlation coefficient of 0.47 between RDI and VCI indicates a moderate positive correlation which means that RDI has stronger association with VCI compared SPEI. This could be because RDI may more directly reflect water availability to plants, thus having a more noticeable impact on vegetation health as measured by VCI. The Pearson correlation results of SPEI and RDI with VCI suggest that meteorological drought affect vegetation cover but these indices do not solely determine it. Other factors such as soil moisture and soil types, irrigation practices, and crop types also contribute to the VCI. This emphasize on the use of multiple indices to get a comprehensive understanding of drought impacts.

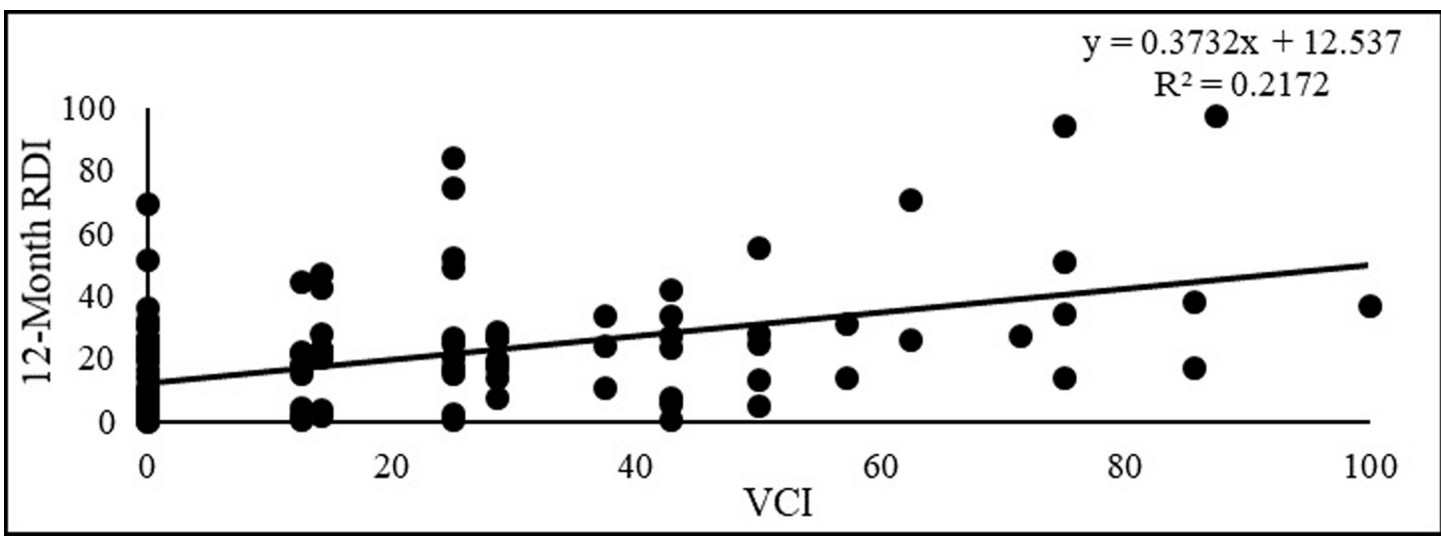

**Fig 7. Scatterplot showing the relationship between VCI and RDI.**

## Discussion

Analysis of drought using the 12-month RDI and SPEI series highlighted a high frequency of severe to extreme droughts in most of the meteorological stations. It's worth noting that the majority of these stations in Baluchistan are situated in arid to semi-arid regions.

The study quantified the short-term variation and frequency of drought in the study area using 1, 3, and 6-month RDI. Results revealed severe to extreme droughts in most parts of the province during 1991, 1998–2000, 2002, 2004, 2010, 2014, and 2018. Similar drought pattern has been observed in other studies conducted in Pakistan [57,58]. Regarding long-term SPEI analysis, the hyper-arid region of Dalbandin emerged as the most affected station by extreme

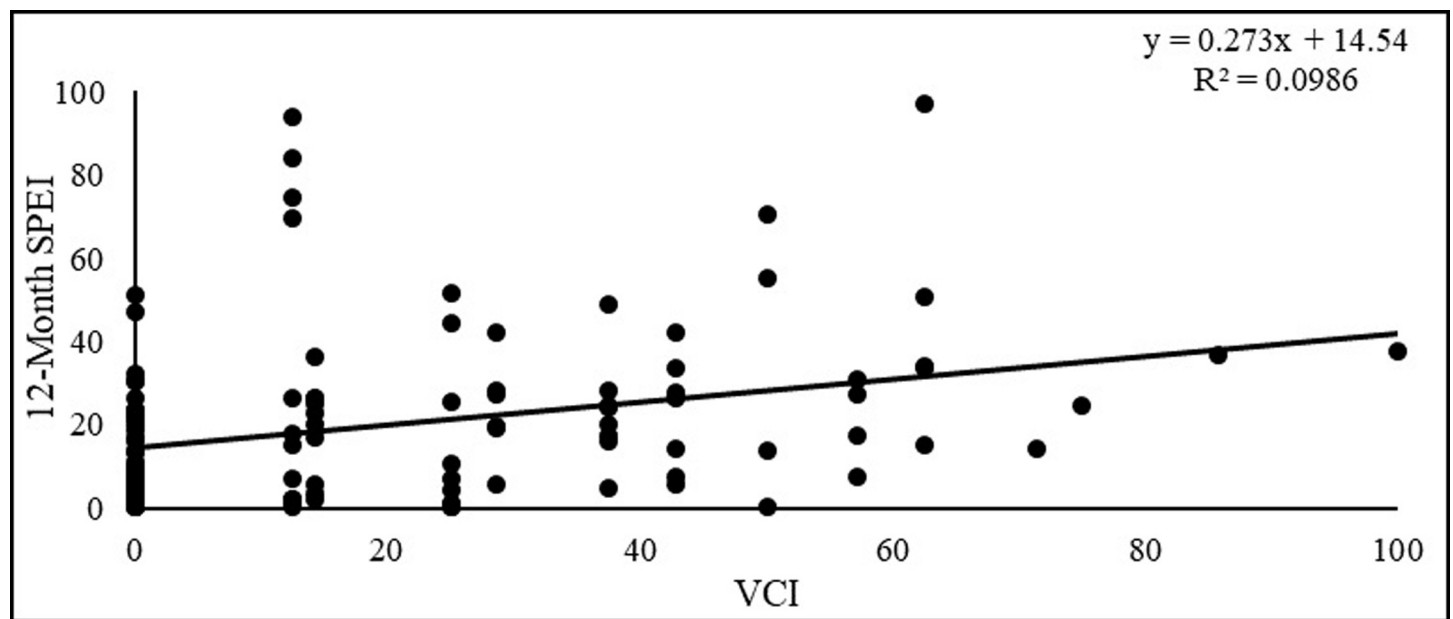

**Fig 8. Scatterplot showing the relationship between VCI and SPEI.**

**Table 10. Pearson correlation matrix.**

| Drought Indices | SPEI | RDI | VCI |
|---|---:|---:|---:|
| SPEI | 1 | 0.74 | 0.31 |
| RDI | 0.74 | 1 | 0.47 |
| VCI | 0.31 | 0.47 | 1 |

drought in the 12-month RDI. Specifically, Dalbandin experienced severe to extreme droughts in 2000, 2002, and 2018, while Lasbella and Khuzdar also faced severe to extreme droughts in 2000, 2002, and 2004. Lasbella exhibited the highest frequency of severe to extreme droughts (8.82%), followed by Dalbandin (8.34%) and Khuzdar (8.33%). Previous research also found severe drought conditions at different stations in Baluchistan during various periods, such as Dalbandin and Quetta (1999–2000), Barkhan, Dalbandin, Lasbella, and Sibi (2002–2003), Zhob (2010–2011), Kalat and Khuzdar (2014–2015), and Panjgur (2017–2018) [29]. The results of this study were further evaluated to identify seasonal drought patterns. It was observed that drought occurs in all seasons, but is more prevalent in the summer and autumn seasons. This elucidates that the drought is primarily due to variations in rainfall in the study area. The droughts in summer season are having more severe impacts on agricultural livelihood and eco-system in the province [59]. Researchers has documented fluctuation and deficit in monsoonal rainfall in Pakistan [41,57,60] and the country is considered among the most vulnerable coun-tries to climate extremes like droughts in South Asia [33]. The recurring droughts in the region are primarily attributed to ENSO, which diminishes precipitation and increase the region's sus-ceptibility to drought hazards as observed in previous studies that highlighted severe drought occurrences in Baluchistan during specific periods, including 1991–1993, 2000–2004, 2014–2015, and 2017–2018, across different meteorological stations [29,41,57,61].

The SPEI results revealed severe to extreme droughts occurring in 1986, 1993, 1998–2002, 2004, 2006, 2014, 2018, 2019, and 2021 across most parts of the province. According to previous research [31], Baluchistan, identified as one of the most drought-prone provinces in Pakistan, has experienced severe droughts during various periods, including 1967–1969, 1971, 1973–1975, 1994, 1998–2002, and 2009–2015 [62]. Barkhan exhibited the highest frequency of severe to extreme droughts, accounting for 15.38% of the years experiencing these conditions. Similar findings from [34] also highlight Barkhan station's frequent occurrence of extreme to severe drought events. Following Barkhan, Kalat and Khuzdar recorded a frequency of 5.88%, with Sibi closely behind at 5.56%. Various studies on SPEI have identified drought spells during 1998–2004, 2006, and 2014–2018 in other regions of Pakistan and in Asia [21,60,63]. These prolonged drought periods, such as the 1998–2002 drought, have been reported to significantly impact human life, agriculture, livestock, and water resources in the affected areas [40]. The arable land of Baluchistan experienced a 50–80% loss due to drought during 1998–2002, resulting in a 50% decrease in agricultural productivity [39,59]. Climate change has led to variations in precipita-tion, which have had severe impacts on drought in Pakistan [64]. Various researchers have reported a decrease in monsoonal rainfall in the western and southern parts of Pakistan, includ-ing Baluchistan and Sindh, making these areas more vulnerable to drought hazards [31,65,66].

The results of the Vegetation Condition Index indicate that the Baluchistan province exhib-ited high spatio-temporal variability in vegetation productivity during the growing seasons over the 23-year study period. Severe to extreme drought conditions were notably observed in 2000, 2001, 2006, and 2010. Among these, the peak severity occurred in 2000 and 2001, affect-ing approximately 69% of the study region. Similar findings were reported by [67], who identi-fied drought events in Pakistan during 2001, 2002, and 2006 based on seasonal measurements of CWSI, NDVI, VHI, and TVDI, confirming a severe drought in 2001, which persisted and

worsened in subsequent years. The significant disturbance in vegetation observed during these periods is attributed to a combination of inadequate vegetation management practices and substantial anthropogenic activities. However, as noted by [68], the impact of human and natural variables on plant life may have attenuated in recent years, potentially influencing the consistency of these effects.

The correlation analysis showed that RDI exhibited a significantly higher correlation value (0.37) compared to SPEI (0.27) in the comparative drought analysis with VCI. These findings suggest that RDI is a more sensitive indicator of agricultural drought conditions. Another study has also demonstrated that the RDI and SPEI indices outperform other drought indices in explaining crop yield anomalies, particularly for crops like cotton and groundnut [69].

The study revealed that Baluchistan is likely to experience more drought spells in the future, which could have a significant impact on meteorological and agricultural systems. These findings about the trend and drought pattern are consistent with a previous study conducted by [34]. The increasing global temperature due to climate change and unpredictable precipitation patterns pose a major challenge in Baluchistan, which already struggles with water scarcity. Since both the RDI and SPEI indices are based on precipitation, temperature, and potential evapotranspiration, they are more reliable for predicting agricultural drought.

## Conclusion

Multiple drought indices, including meteorological drought indices (RDI, SPEI), and the vegetation drought index (VCI) were utilized to quantify the frequency, spatiotemporal distribution, and correlation among these indices in Baluchistan. Severe to extreme drought occurred in most stations in the 1, 3, 6, and 12-month RDI in 1988, 1991, 1998–2002, 2004, 2010, 2018, and 2019 in the study area. Lasbella station had the highest frequency of severe to extreme droughts in the 12-month RDI, at 8.82%, followed by Dalbandin at 8.34%. The 1, 3, 6, and 12-month SPEI showed severe to extreme drought in the years 1986, 1993, 2000–2002, 2004, 2010, 2018, 2019, and 2021 in most parts of the study region. The Barkhan station had the highest frequency of severe to extreme droughts in the 12-month SPEI, with 15.38%, followed by Khuzdar at 5.88%. The results of the vegetation drought index (VCI) showed that severe to extreme drought occurred during 2000, 2001, 2006, and 2010. The highest percentage of drought was noted in 2000, 2001, affecting 69% of the study region. The correlation coefficient results between the meteorological (RDI, SPEI) and agricultural drought (VCI) indices showed a positive relationship. This means that as RDI and SPEI increase, VCI also increases. In other words, more severe agricultural droughts are associated with higher values of RDI and SPEI. The study recommends that the RDI and SPEI drought indices are useful for assessing and managing agricultural drought.

## Acknowledgments

The authors acknowledge Pakistan Meteorological Department (PMD) for providing the climate data for this research.

## Author Contributions

**Conceptualization:** Muhammad Rafiq, Aun Zahoor.

**Data curation:** Muhammad Rafiq, Kamil Khan.

**Formal analysis:** Muhammad Rafiq, Ghani Rahman, Khawar Sohail, Aun Zahoor, Farrukh Gujjar.

**Funding acquisition:** Yue Cong Li.

**Methodology:** Muhammad Rafiq.

**Resources:** Muhammad Rafiq.

**Software:** Muhammad Rafiq, Ghani Rahman.

**Supervision:** Yue Cong Li.

**Visualization:** Ghani Rahman.

**Writing – original draft:** Muhammad Rafiq.

**Writing – review & editing:** Hyun-Han Kwon.

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
