## [Decision Letter · Decision Letter 0]

30 Apr 2024

PONE-D-24-09924Regional Characterization of Meteorological and Agricultural Drought in Baluchistan Province, PakistanPLOS ONE

Dear Dr. Muhammad Rafiq,

Thank you for submitting your manuscript to PLOS ONE. After careful consideration, we feel that it has merit but does not fully meet PLOS ONE’s publication criteria as it currently stands. Therefore, we invite you to submit a revised version of the manuscript that addresses the points raised during the review process. Please submit your revised manuscript by Jun 14 2024 11:59PM. If you will need more time than this to complete your revisions, please reply to this message or contact the journal office at plosone@plos.org. Please include the following items when submitting your revised manuscript:A rebuttal letter that responds to each point raised by the academic editor and reviewer(s). You should upload this letter as a separate file labeled 'Response to Reviewers'.A marked-up copy of your manuscript that highlights changes made to the original version. You should upload this as a separate file labeled 'Revised Manuscript with Track Changes'.An unmarked version of your revised paper without tracked changes. You should upload this as a separate file labeled 'Manuscript'.

We look forward to receiving your revised manuscript.

Kind regards,

Salim Heddam

Academic Editor

PLOS ONE

Journal Requirements:

4. For studies involving third-party data, we encourage authors to share any data specific to their analyses that they can legally distribute. PLOS recognizes, however, that authors may be using third-party data they do not have the rights to share. When third-party data cannot be publicly shared, authors must provide all information necessary for interested researchers to apply to gain access to the data. (https://journals.plos.org/plosone/s/data-availability#loc-acceptable-data-access-restrictions) 

6. We note that Figure 1 and 6 in your submission contain map/satellite images which may be copyrighted. All PLOS content is published under the Creative Commons Attribution License (CC BY 4.0), which means that the manuscript, images, and Supporting Information files will be freely available online, and any third party is permitted to access, download, copy, distribute, and use these materials in any way, even commercially, with proper attribution. For these reasons, we cannot publish previously copyrighted maps or satellite images created using proprietary data, such as Google software (Google Maps, Street View, and Earth). For more information, see our copyright guidelines: http://journals.plos.org/plosone/s/licenses-and-copyright.

a. You may seek permission from the original copyright holder of Figure 1 and 6 to publish the content specifically under the CC BY 4.0 license.  

7. Please ensure that you refer to Figure 1 in your text as, if accepted, production will need this reference to link the reader to the figure.

8. We note you have included a table to which you do not refer in the text of your manuscript. Please ensure that you refer to Table 3 in your text; if accepted, production will need this reference to link the reader to the Table.

Additional Editor Comments:

Reviewer 1#:In order to identify drought episodes in Baluchistan, a comprehensive analysis was conducted using three key indices: the Reconnaissance Drought Index (RDI), Standardized Precipitation Evapotranspiration Index (SPEI), and Vegetation Condition Index (VCI). This analysis took into account various factors such as precipitation, temperature, potential evapotranspiration (PET), and vegetation condition. The work is original, but I recommend that the following major changes be made before it is accepted for publication.

Why the selected models were chosen and their advantages should be expressed.

My literature section is limited, sources from 2023 and 2024 should be added.

After its first mention, vegetation drought index (VCI) should be abbreviated

In order to provide a more comprehensive literature section, it is suggested to incorporate the following paper.

• Comparison of meteorological indices for drought monitoring and evaluating: a case study from Euphrates basin, Turkey

It is necessary to verify all equations. Equation 8 contains several missing and erroneous statements that need to be addressed.

Here is an explanation of how the spatio-temporal distribution of drought was formed. To enhance the method section, it is advisable to employ the state interpolation method and provide a comprehensive explanation of the interpolation technique.

Please cite fallowing paper fort his aim:

• Spatio-temporal analysis of meteorological and hydrological droughts in the Euphrates Basin, Turkey

• Spatial analysis of seasonal precipitation using various interpolation methods in the Euphrates basin, Turkey

In the introduction section, it is important to highlight the advantages of interpolation methods. Interpolation methods offer several benefits that make them valuable in various fields. Firstly, they allow for the estimation of values between known data points, providing a continuous representation of the data. This helps in accurately predicting unknown values and filling in gaps in the data set. Additionally, interpolation methods aid in reducing noise and smoothing out irregularities in the data, leading to a more precise and reliable analysis. Furthermore, these methods are computationally efficient and can handle large datasets efficiently. By incorporating interpolation methods into the introduction section, readers will gain a deeper understanding of the significance and usefulness of these techniques.

The discussion section is not deep. It would be good to improve.

Suggestions for future studies can be made and it can be suggested which other water resources problems the wavelet transform can be used to solve.

Reviewer 2#:The research entitled “Regional Characterization of Meteorological and Agricultural Drought in Baluchistan Province, Pakistan” aimed to examine the spatiotemporal pattern of three competitive drought indices (RDI, SPEI, and VCI) in Balochistan province of Pakistan.

The study is important to reveal the regional climatic characteristics of largest province of Pakistan and related to its vegetative (Agricultural) phenomenon. Overall, research design is valid but substantial improvements are needed. My specific comments are:

1. Although sample stations are good representative of study area, but the methods are very simple and presents less novelty. Research design need to be enhanced adding drought frequency and trend analysis. Follow the drought run theory.

2. How VCI is calculated? Which remote sensing dataset used? The method is not properly described. Which temporal scale is used? If it is calculated using Landsat, the results might be uncertain. MODIS is more recommended for a province level study area and will provide more consistent results. In this case, VCI can be regressed with 20 years analysis (based on the dates of datasets). At what temporal scale VCI is generated? 12 months? Clarify the methodology.

3. Spatial presentation of RDI and SPEI can be added for spatial comparison with VCI at same scale.

4. A Pearson correlation matrix can be added for all indices for more clear statistical relationship between the variables. Regression scatterplot reveals a very weak relationship of VCI with SPEI. Need to be discussed, why? No need of table 8

5. Discussion part need to be more strong with more relevant reference and discussing the major causes of previous drought events.

Overall, research design of the study need to be enhanced adding more sound methods and results.

Reviewers' comments:

Reviewer's Responses to Questions

**Comments to the Author**

1. Is the manuscript technically sound, and do the data support the conclusions?

Reviewer #1: Yes

Reviewer #2: Yes

2. Has the statistical analysis been performed appropriately and rigorously? 

Reviewer #1: Yes

Reviewer #2: Yes

3. Have the authors made all data underlying the findings in their manuscript fully available?

Reviewer #1: Yes

Reviewer #2: No

4. Is the manuscript presented in an intelligible fashion and written in standard English?

Reviewer #1: Yes

Reviewer #2: Yes

5. Review Comments to the Author

Reviewer #1: In order to identify drought episodes in Baluchistan, a comprehensive analysis was conducted using three key indices: the Reconnaissance Drought Index (RDI), Standardized Precipitation Evapotranspiration Index (SPEI), and Vegetation Condition Index (VCI). This analysis took into account various factors such as precipitation, temperature, potential evapotranspiration (PET), and vegetation condition. The work is original, but I recommend that the following major changes be made before it is accepted for publication.

Why the selected models were chosen and their advantages should be expressed.

My literature section is limited, sources from 2023 and 2024 should be added.

After its first mention, vegetation drought index (VCI) should be abbreviated

In order to provide a more comprehensive literature section, it is suggested to incorporate the following paper.

• Comparison of meteorological indices for drought monitoring and evaluating: a case study from Euphrates basin, Turkey

It is necessary to verify all equations. Equation 8 contains several missing and erroneous statements that need to be addressed.

Here is an explanation of how the spatio-temporal distribution of drought was formed. To enhance the method section, it is advisable to employ the state interpolation method and provide a comprehensive explanation of the interpolation technique.

Please cite fallowing paper fort his aim:

• Spatio-temporal analysis of meteorological and hydrological droughts in the Euphrates Basin, Turkey

• Spatial analysis of seasonal precipitation using various interpolation methods in the Euphrates basin, Turkey

In the introduction section, it is important to highlight the advantages of interpolation methods. Interpolation methods offer several benefits that make them valuable in various fields. Firstly, they allow for the estimation of values between known data points, providing a continuous representation of the data. This helps in accurately predicting unknown values and filling in gaps in the data set. Additionally, interpolation methods aid in reducing noise and smoothing out irregularities in the data, leading to a more precise and reliable analysis. Furthermore, these methods are computationally efficient and can handle large datasets efficiently. By incorporating interpolation methods into the introduction section, readers will gain a deeper understanding of the significance and usefulness of these techniques.

The discussion section is not deep. It would be good to improve.

Suggestions for future studies can be made and it can be suggested which other water resources problems the wavelet transform can be used to solve.

Reviewer #2: The research entitled “Regional Characterization of Meteorological and Agricultural Drought in Baluchistan Province, Pakistan” aimed to examine the spatiotemporal pattern of three competitive drought indices (RDI, SPEI, and VCI) in Balochistan province of Pakistan.

The study is important to reveal the regional climatic characteristics of largest province of Pakistan and related to its vegetative (Agricultural) phenomenon. Overall, research design is valid but substantial improvements are needed. My specific comments are:

1. Although sample stations are good representative of study area, but the methods are very simple and presents less novelty. Research design need to be enhanced adding drought frequency and trend analysis. Follow the drought run theory.

2. How VCI is calculated? Which remote sensing dataset used? The method is not properly described. Which temporal scale is used? If it is calculated using Landsat, the results might be uncertain. MODIS is more recommended for a province level study area and will provide more consistent results. In this case, VCI can be regressed with 20 years analysis (based on the dates of datasets). At what temporal scale VCI is generated? 12 months? Clarify the methodology.

3. Spatial presentation of RDI and SPEI can be added for spatial comparison with VCI at same scale.

4. A Pearson correlation matrix can be added for all indices for more clear statistical relationship between the variables. Regression scatterplot reveals a very weak relationship of VCI with SPEI. Need to be discussed, why? No need of table 8

5. Discussion part need to be more strong with more relevant reference and discussing the major causes of previous drought events.

Overall, research design of the study need to be enhanced adding more sound methods and results.

6. PLOS authors have the option to publish the peer review history of their article (what does this mean?). If published, this will include your full peer review and any attached files.

Reviewer #1: No

Reviewer #2: **Yes: **Sana Arshad

---

## [Author Response · Author response to Decision Letter 0]

6 Jun 2024

Regional Characterization of Meteorological and Agricultural Drought in Baluchistan Province, Pakistan 

Point-to-Point Response to the Reviewer’s Comments

We are thankful to the editor and team for their timely response and efforts. We are also thankful to reviewers for their deep and valuable comments that improved the quality of the manuscript. We are submitting our point-by-point response to the comments and an updated manuscript file with highlighted in red colour. We are hopeful that the revised manuscript has improved to the level of satisfaction and approval. We also hope that now the manuscript meets the journal’s publication requirements. Number-wise answers to their specific comments and suggestions are as follows. ________________________________________

Editor Comments: 

Authors response: We highly appreciate your effort. We carefully checked the format and set according to the journal requirements.

Please note that funding information should not appear in any section or other areas of your manuscript. We will only publish funding information present in the Funding Statement section of the online submission form. Please remove any funding-related text from the manuscript.

Authors response: We removed the funding information from the manuscript.

We note that the grant information you provided in the ‘Funding Information’ and ‘Financial Disclosure’ sections do not match.

Authors response: We reconfirmed the provided grant information and correct it carefully.

For studies involving third-party data, we encourage authors to share any data specific to their analyses that they can legally distribute. PLOS recognizes, however, that authors may be using third-party data they do not have the rights to share. When third-party data cannot be publicly shared, authors must provide all information necessary for interested researchers to apply to gain access to the data. (https://journals.plos.org/plosone/s/data-availability#loc-acceptable-data-access-restrictions).

Authors response: No third party data involve in this research.

PLOS requires an ORCID iD for the corresponding author in Editorial Manager on papers submitted after December 6th, 2016.

Authors response: ORCID ID provided for the corresponding author.

We note that Figure 1 and 6 in your submission contain map/satellite images which may be copyrighted.

Authors response: The figure 1 and 6 is solely the authors own maps. There is no copy right issue. These are prepare under Creative Commons Attribution License (CC BY 4.0)." Authors response: The figure 1 and 7 is solely the authors own maps. The Base layers used in Figure 1 are: Digital Elevation Data which is freely available from Website 30-Meter SRTM Elevation Data Downloader (dwtkns.com). The Shape files used in preparing map for Figure 1, 2 and 7 is digitized from the Map available from Survey of Pakistan Website: http://www.surveyofpakistan.gov.pk/Detail/MTUzYWU5ZGItNTA4NS00MDlkLWFlODctNTRkY2JmNWI0Mjg2. In the figure 7 we have used Landsat NDVI data which have been discussed in the methodology section and properly cited the Landsat imageries.

Further online Base map has been removed to avoid any Copy right issue.

Please ensure that you refer to Figure 1 in your text as, if accepted, production will need this reference to link the reader to the figure.

Authors response: Reference to the Figure 1 provided in the text.

We note you have included a table to which you do not refer in the text of your manuscript. Please ensure that you refer to Table 3 in your text; if accepted, production will need this reference to link the reader to the Table.

Authors response: Reference to the Table 3 provided in the text.

Reviewer #1: 

General comments: In order to identify drought episodes in Baluchistan, a comprehensive analysis was conducted using three key indices: the Reconnaissance Drought Index (RDI), Standardized Precipitation Evapotranspiration Index (SPEI), and Vegetation Condition Index (VCI). This analysis took into account various factors such as precipitation, temperature, potential evapotranspiration (PET), and vegetation condition. The work is original, but I recommend that the following major changes be made before it is accepted for publication.:

Authors response: We highly appreciate the effort of the reviewer for the detail review of the manuscript. The reviewer comments and suggestions improved the quality of the manuscript and we authros are highly indebted for your kind words.

Why the selected models were chosen and their advantages should be expressed.

Authors response: We discussed the advantages of each model in the revised introduction section and highlighted the importance of these models that why we selected these. 

My literature section is limited, sources from 2023 and 2024 should be added.

Authors response: We highly appreciate the effort of the reviewer for the detail review of the manuscript. Authors added new citation and literature especially citations from 2023 and 2024.

After its first mention, vegetation drought index (VCI) should be abbreviated.

Authors response: We abbrevaited the Vegetation Drought Index (VCI) in the revised manuscript as suggested by worthy reviewer. 

In order to provide a more comprehensive literature section, it is suggested to incorporate the following paper. Comparison of meteorological indices for drought monitoring and evaluating: a case study from Euphrates basin

Authors response: We highly appreciate the effort of the reviewer for the detail review of the manuscript. We cited the suggested research article in the Introduction Section. 

It is necessary to verify all equations. Equation 8 contains several missing and erroneous statements that need to be addressed. Here is an explanation of how the spatio-temporal distribution of drought was formed. To enhance the method section, it is advisable to employ the state interpolation method and provide a comprehensive explanation of the interpolation technique.

Authors response: We rechecked all the equations and statements relevant to the given equations and corrected. We added IDW interpolation technique in the methodology section and prepared maps in the results.

Please cite fallowing paper fort his aim: • Spatio-temporal analysis of meteorological and hydrological droughts in the Euphrates Basin, Turkey. • Spatial analysis of seasonal precipitation using various interpolation methods in the Euphrates basin, Turkey.

Authors response: We appreciate your efforts. We added the suggested research article in citations to improve the literature section.

In the introduction section, it is important to highlight the advantages of interpolation methods. Interpolation methods offer several benefits that make them valuable in various fields. Firstly, they allow for the estimation of values between known data points, providing a continuous representation of the data. This helps in accurately predicting unknown values and filling in gaps in the data set. Additionally, interpolation methods aid in reducing noise and smoothing out irregularities in the data, leading to a more precise and reliable analysis. Furthermore, these methods are computationally efficient and can handle large datasets efficiently. By incorporating interpolation methods into the introduction section, readers will gain a deeper understanding of the significance and usefulness of these techniques.

Authors response: We enhanced the introduction and methodology sections in light of your valuable suggestions. Thank you for your time and comments.

The discussion section is not deep. It would be good to improve. Suggestions for future studies can be made and it can be suggested which other water resources problems the wavelet transform can be used to solve.

Authors response: We improved the discussion section as well and tried to give future research direction and mitigation strategies. Thank you for your valuable suggestions and time.

Thank you for your valuable comment and time.

Reviewer #2: 

The research entitled “Regional Characterization of Meteorological and Agricultural Drought in Baluchistan Province, Pakistan” aimed to examine the spatiotemporal pattern of three competitive drought indices (RDI, SPEI, and VCI) in Balochistan province of Pakistan.

The study is important to reveal the regional climatic characteristics of largest province of Pakistan and related to its vegetative (Agricultural) phenomenon. Overall, research design is valid but substantial improvements are needed. My specific comments are:.

Authors response: Thank you for your thoughtful comments regarding our paper. We are pleased to hear that you consider our study importance. Your input is invaluable in improving the quality and impact of our research paper.

1. Although sample stations are good representative of study area, but the methods are very simple and presents less novelty. Research design need to be enhanced adding drought frequency and trend analysis. Follow the drought run theory.

Authors response: Thank you for your valuable feedback. We added the drought frequency and trend analysis following the droughy run theory. The figures are redesigned in R based on the output of run theory that shows the freuency and trend of each drought category with different colors. We also added frequency table for further explanation.

2. How VCI is calculated? Which remote sensing dataset used? The method is not properly described. Which temporal scale is used? If it is calculated using Landsat, the results might be uncertain. MODIS is more recommended for a province level study area and will provide more consistent results. In this case, VCI can be regressed with 20 years analysis (based on the dates of datasets). At what temporal scale VCI is generated? 12 months? Clarify the methodology.

Authors response: We improved the methodology section for VCI calculation and explained the datasets used. We calculated VCI based on Landsat NDVI datasets derived from Google Earth Engine. For an area like Baluchistan, Landsat provides more accurate results as the area remains cloud-free, and the data is available for a more extended period than MODIS. VCI is calculated from April to July, taking the average value of this period since, at this time, most of Baluchistan is covered with green vegetation. Therefore, we selected this period for VCI analysis.

3. Spatial presentation of RDI and SPEI can be added for spatial comparison with VCI at same scale..

Authors response: We appreciate your time and valuable comments. Considering the paper's requirements, we believe it is already quite lengthy, so we decided not to add further maps, as this would increase the length significantly. We hope you understand our decision to skip this valuable suggestion. We acknowledge that it would enhance the quality of the paper, but we apologize for not including it due to the paper's current length. 

4. A Pearson correlation matrix can be added for all indices for more clear statistical relationship between the variables. Regression scatterplot reveals a very weak relationship of VCI with SPEI. Need to be discussed, why? 

Authors response: We added the Pearson Correlation Matric in the revised manuscript and discuss the possible reason of low correlation of SPEI and RDI with VCI.

5. Discussion part need to be more strong with more relevant reference and discussing the major causes of previous drought events. Overall, research design of the study need to be enhanced adding more sound methods and results..

Authors response: Thank you for your constructive feedback. We added further discussion specifically on the causes of previous droughts in the study area. We thoroughly revised the manuscript and hopefully you will find it in more improved form. 

Thank you for your valuable comment and time.

---

## [Decision Letter · Decision Letter 1]

19 Jun 2024

PONE-D-24-09924R1Regional characterization of meteorological and agricultural drought in Baluchistan province, PakistanPLOS ONE

Dear Dr. Rahman,

Thank you for submitting your manuscript to PLOS ONE. After careful consideration, we feel that it has merit but does not fully meet PLOS ONE’s publication criteria as it currently stands. Therefore, we invite you to submit a revised version of the manuscript that addresses the points raised during the review process. Please submit your revised manuscript by Aug 03 2024 11:59PM. If you will need more time than this to complete your revisions, please reply to this message or contact the journal office at plosone@plos.org. Please include the following items when submitting your revised manuscript:A rebuttal letter that responds to each point raised by the academic editor and reviewer(s). You should upload this letter as a separate file labeled 'Response to Reviewers'.A marked-up copy of your manuscript that highlights changes made to the original version. You should upload this as a separate file labeled 'Revised Manuscript with Track Changes'.An unmarked version of your revised paper without tracked changes. You should upload this as a separate file labeled 'Manuscript'.If applicable, we recommend that you deposit your laboratory protocols in protocols.io to enhance the reproducibility of your results. Protocols.io assigns your protocol its own identifier (DOI) so that it can be cited independently in the future. For instructions see: https://journals.plos.org/plosone/s/submission-guidelines#loc-laboratory-protocols. Additionally, PLOS ONE offers an option for publishing peer-reviewed Lab Protocol articles, which describe protocols hosted on protocols.io. Read more information on sharing protocols at https://plos.org/protocols?utm_medium=editorial-email&utm_source=authorletters&utm_campaign=protocols.

We look forward to receiving your revised manuscript.

Kind regards,

Salim Heddam

Academic Editor

PLOS ONE

Journal Requirements:

Additional Editor Comments:

Reviewer 1#:

All revisions were appied

Can accept Paper after the font size of the letters in Figure 3 is enlarged.

Reviewer 2#:

Although authors have made substantial changes to the manuscript and addressed the comments. Still, a few parts need to be improved

1. Introduction: provide a few more recent drought example studies from Pakistan.

2. The resolution of newly added figures is not good. Please improve the presentation of figure 3, 4, 5 and 6. Increase the legend and label size and try to concise them to take less space. Captains need to be revised. Remove the word “spatio” from these titles.

3. Present the statistical analysis with a good heading. Replace “Correlation statistical tool” with term “statistical analysis through Pearson correlation coefficient and linear regression”.

4. What about Mann-Kendall and Sen’s slope analysis? Why not performed?

5. Align all figures equally in manuscript.

Reviewers' comments:

Reviewer's Responses to Questions

**Comments to the Author**

1. If the authors have adequately addressed your comments raised in a previous round of review and you feel that this manuscript is now acceptable for publication, you may indicate that here to bypass the “Comments to the Author” section, enter your conflict of interest statement in the “Confidential to Editor” section, and submit your "Accept" recommendation.

Reviewer #1: All comments have been addressed

Reviewer #2: All comments have been addressed

2. Is the manuscript technically sound, and do the data support the conclusions?

Reviewer #1: Yes

Reviewer #2: Yes

3. Has the statistical analysis been performed appropriately and rigorously? 

Reviewer #1: Yes

Reviewer #2: Yes

4. Have the authors made all data underlying the findings in their manuscript fully available?

Reviewer #1: Yes

Reviewer #2: No

5. Is the manuscript presented in an intelligible fashion and written in standard English?

Reviewer #1: Yes

Reviewer #2: Yes

6. Review Comments to the Author

Reviewer #1: All revisions were appied

Can accept Paper after the font size of the letters in Figure 3 is enlarged.

Reviewer #2: Although authors have made substantial changes to the manuscript and addressed the comments. Still, a few parts need to be improved

1. Introduction: provide a few more recent drought example studies from Pakistan.

2. The resolution of newly added figures is not good. Please improve the presentation of figure 3, 4, 5 and 6. Increase the legend and label size and try to concise them to take less space. Captains need to be revised. Remove the word “spatio” from these titles.

3. Present the statistical analysis with a good heading. Replace “Correlation statistical tool” with term “statistical analysis through Pearson correlation coefficient and linear regression”.

4. What about Mann-Kendall and Sen’s slope analysis? Why not performed?

5. Align all figures equally in manuscript.

7. PLOS authors have the option to publish the peer review history of their article (what does this mean?). If published, this will include your full peer review and any attached files.

Reviewer #1: No

Reviewer #2: No

---

## [Author Response · Author response to Decision Letter 1]

21 Jun 2024

Research: Regional characterization of meteorological and agricultural drought in Baluchistan province, Pakistan

We revised the manuscript according to the directions of the editor and reviewers.

Journal Requirements:

Author Response:

We carefully checked all the references in the list on Retraction Watch, and based on the information obtained, none of the references have been identified as retracted.

Reviewer 1:

Can accept Paper after the font size of the letters in Figure 3 is enlarged.

Author Response: We revised the figures and enlarged the font size.

Reviewer 2: 

1. Introduction: provide a few more recent drought example studies from Pakistan.

Author Response: We included the latest literature and examples from Pakistan in both the Introduction and Discussion Sections.

2. The resolution of newly added figures is not good. Please improve the presentation of figure 3, 4, 5 and 6. Increase the legend and label size and try to concise them to take less space. Captains need to be revised. Remove the word “spatio” from these titles.

Author Response: We improved the presentation of the mentioned figures and increased the font size used in the figures. We condensed the figures to minimize the occupied space.

3. Present the statistical analysis with a good heading. Replace “Correlation statistical tool” with term “statistical analysis through Pearson correlation coefficient and linear regression”.

Author Response: We revised the heading as suggested by the reviewer.

4. What about Mann-Kendall and Sen’s slope analysis? Why not performed?

Author Response: The focus of this research was on drought, so we did not use any trend statistics in this research. 

5. Align all figures equally in manuscript.

Author Response: We carefully checked and aligned all the figures.

---

## [Decision Letter · Decision Letter 2]

2 Jul 2024

Regional characterization of meteorological and agricultural drought in Baluchistan province, Pakistan

PONE-D-24-09924R2

Dear Dr. Rahman

We’re pleased to inform you that your manuscript has been judged scientifically suitable for publication and will be formally accepted for publication once it meets all outstanding technical requirements.

Kind regards,

Salim Heddam

Academic Editor

PLOS ONE

Additional Editor Comments (optional):

Reviewer 1#:The article is adequately developed. figures and enlarged the font size looks well. Thanks alot for taking attention.

Reviewer 2#:I have now reviewed the research titled "Regional characterization of meteorological and agricultural drought in Baluchistan province, Pakistan.

Authors have improved the manuscript incorporating all suggestions.

I recommend for acceptance

Reviewers' comments:

Reviewer's Responses to Questions

**Comments to the Author**

1. If the authors have adequately addressed your comments raised in a previous round of review and you feel that this manuscript is now acceptable for publication, you may indicate that here to bypass the “Comments to the Author” section, enter your conflict of interest statement in the “Confidential to Editor” section, and submit your "Accept" recommendation.

Reviewer #1: All comments have been addressed

Reviewer #2: (No Response)

2. Is the manuscript technically sound, and do the data support the conclusions?

Reviewer #1: Yes

Reviewer #2: (No Response)

3. Has the statistical analysis been performed appropriately and rigorously? 

Reviewer #1: Yes

Reviewer #2: (No Response)

4. Have the authors made all data underlying the findings in their manuscript fully available?

Reviewer #1: Yes

Reviewer #2: (No Response)

5. Is the manuscript presented in an intelligible fashion and written in standard English?

Reviewer #1: Yes

Reviewer #2: (No Response)

6. Review Comments to the Author

Reviewer #1: The article is adequately developed. figures and enlarged the font size looks well. Thanks alot for taking attention.

Reviewer #2: I have now reviewed the research titled "Regional characterization of meteorological and agricultural drought in Baluchistan province, Pakistan. Authors have improved the manuscript incorporating all suggestions.

I recommend for acceptance

7. PLOS authors have the option to publish the peer review history of their article (what does this mean?). If published, this will include your full peer review and any attached files.

Reviewer #1: No

Reviewer #2: No

---

## [Editor Report · Acceptance letter]

9 Aug 2024

PONE-D-24-09924R2 

PLOS ONE

Dear Dr. Rahman, 

I'm pleased to inform you that your manuscript has been deemed suitable for publication in PLOS ONE. Congratulations! Your manuscript is now being handed over to our production team.

Kind regards, 

on behalf of

Dr. Salim Heddam 

Academic Editor

PLOS ONE